# Association between gabapentinoid treatment, concurrent use with opioid or benzodiazepine and the risk of drug poisoning: A self-controlled case series study

Andrew S. C. Yuen[1,2], Boqing Chen[1,2], Adrienne Y. L. Chan[3,4], Joseph F. Hayes[5,6], David P. J. Osborn[5,6], Frank M. C. Besag[1,7,8], Wallis C. Y. Lau[1,2,4], Ian C. K. Wong[3,4,9], Li Wei[1,2], Kenneth K. C. Man[1,2,4]*

1 Research Department of Practice and Policy, School of Pharmacy, University College London, London, United Kingdom, 2 Centre for Medicines Optimisation Research and Education, University College London Hospitals NHS Foundation Trust, London, United Kingdom, 3 School of Pharmacy, Aston University, Birmingham, United Kingdom, 4 Centre for Safe Medication Practice and Research, Department of Pharmacology and Pharmacy, Li Ka Shing Faculty of Medicine, University of Hong Kong, Pok Fu Lam, Hong Kong SAR, 5 Division of Psychiatry, University College London, London, United Kingdom, 6 North London NHS Foundation Trust, London, United Kingdom, 7 East London Foundation NHS Trust, Bedfordshire, London, United Kingdom, 8 Institute of Psychiatry, Psychology and Neuroscience, King's College London, London, United Kingdom, 9 School of Pharmacy, Medical Sciences Division, Macau University of Science and Technology, Taipa, Macau

* kenneth.man@ucl.ac.uk

## Abstract

### Background

Consumption of gabapentinoids has increased worldwide in recent years, and the association between its use and drug poisoning is of public health concern. This study aimed to investigate the association between gabapentinoid treatment and the risk of drug poisoning.

### Methods and findings

In this within-individual study, we utilised data from the United Kingdom (UK) Clinical Practice Research Datalink (CPRD) Aurum database linked to the Hospital Episode Statistics (HES) and Office for National Statistics (ONS). The analysis included individuals aged 18 or above who were prescribed gabapentinoids and had an incident all-cause drug poisoning event between 1st January 2010 and 31st December 2020. Using the self-controlled case series (SCCS) design, we assessed the risk of drug poisoning incidence in predefined risk periods: 90 days before treatment initiation, first 28, 29–56, 57–84 days, and the remaining treatment time. Concomitant use with opioids/benzodiazepines was also evaluated. Adjusted incidence rate ratios (aIRRs) were calculated using conditional Poisson regression. A case-case-time-control (CCTC) analysis was also conducted, with adjusted odds ratio (aOR)

**Data availability statement:** This study is based on data from the Clinical Practice Research Datalink (CPRD) obtained under licence from the UK Medicines and Healthcare products Regulatory Agency (MHRA). The data is provided by patients and collected by the National Health Service (NHS) as part of their care and support. The interpretation and conclusions contained in this study are those of the author/s alone. The SAS code of this study is made available at https://github.com/andrewyuen97/SCCS_drug_poisoning. The Zenodo URL is https://doi.org/10.5281/zenodo.19105047. Due to the data user agreement between UCL and CPRD, researchers are not authorised to share CPRD data. Access to CPRD data, including UK Primary Care Data, and linked data such as Hospital Episode Statistics (HES) and Office for National Statistics (ONS), is subject to protocol approval via CPRD's Research Data Governance (RDG) Process, see https://cprd.com/data-access for further details.

**Funding:** ASCY and KKCM received a grant from NIHR UCLH Biomedical Research Centre (https://www.uclhospitals.brc.nihr.ac.uk) to support the Patient and Public Involvement activities related to this submission. The grant number is BRC1141/PPI/SY/104990. The funding organisations had no role in the study design, execution and analysis, and manuscript conception, planning, writing and decision to publish.

**Competing interests:** I have read the journal's policy and the authors of this manuscript have the following competing interests: ASCY reports grant from the University College London Hospitals NIHR Biomedical Research Centre, College of Mental Health Pharmacy, and UK Turing Scheme. AYLC reports grant from the AIR@innoHK programme of the Hong Kong Innovation and Technology Commission, consulting fees from ADHD Europe, outside of the submitted work. JFH reports grants from UK Research and Innovation grant and the University College London Hospitals NIHR Biomedical Research Centre and the NIHR North Thames Applied Research Collaboration. He received consulting fees from the Wellcome Trust, Juli Health and Swiss Re. DPJO reports

calculated to validate the findings from the main SCCS analysis. All analyses have adjusted for key time-varying confounders, including age, season, and concomitant use of opioids, antiseizure medications, psychotropic medications, and non-steroidal anti-inflammatory drugs (NSAIDs).

16,827 individuals met the inclusion criteria and were included in the SCCS analysis. The risk of drug poisoning, compared with the reference periods, increased during the first 28 days of gabapentinoid treatment (aIRR = 1.81, 95% confidence interval [CI] [1.66, 1.99]; $p < 0.001$), eventually dropped to 1.11 (95% CI [1.05, 1.17]; $p < 0.001$) in the remainder of the treatment period. Notably, the risk was doubled during the 90-day preceding treatment initiation (aIRR = 2.09, 95% CI [1.98, 2.21]; $p < 0.001$). Co-administration with opioids elevated the risk by 30%, while benzodiazepines increased it 2-fold. The CCTC analysis also detected an increased aOR of 1.36 (95% CI [1.12, 1.65]; $p = 0.002$) of receiving gabapentinoid treatment within 30 days prior to a drug poisoning event. The SCCS approach cannot completely exclude the effect of unmeasured time-varying confounders, such as transient changes in socioeconomic status, major life events, or illicit drug use, although the negative control analysis did not suggest meaningful residual confounding.

## Conclusions

The results suggest that gabapentinoid is associated with an increased risk of drug poisoning. Close monitoring throughout gabapentinoid treatment journey for drug poisoning is needed, especially at the initial phase. Concomitant use with opioid or benzodiazepines should be avoided.

## Author summary
### Why was this study done?

- Use of gabapentinoids, which are indicated for neuropathic pain, epilepsy, and generalised anxiety disorder, and are also used off label for fibromyalgia, sleep disorders, and other chronic pain conditions, has increased substantially over the past decade. However, there are growing concerns that they may contribute to drug poisoning and overdose, particularly when combined with opioids or benzodiazepines.

- Previous studies have mainly focussed on high-risk populations or opioid users, and only one has examined unintentional overdose, with important limitations such as inadequate adjustment for concomitant medications.

- There is a need to characterise how the risk of drug poisoning changes before and after starting gabapentinoids, and during concomitant opioid/ benzodiazepine use, to provide clinicians with robust evidence to guide safer prescribing and monitoring.

grants from the University College London Hospitals NIHR Biomedical Research Centre and the NIHR North Thames Applied Research Collaboration. WCYL reports grants from Diabetes UK, AIR@InnoHK of the Hong Kong Innovation and Technology Commission. ICKW reports research funding from the Hong Kong Research Grants Council, the Hong Kong Health and Medical Research Fund, the National Institute for Health Research in England, the European Commission, IQVIA, Amgen and GSK, consulting fees from IQVIA and World Health Organization, Advance Data Analytics for Medical Science (ADAMS) Limited in Hong Kong outside of the submitted work. He is a non-executive director of Jacobson Pharma Corporation Limited, Advance Data Analytics for Medical Science (ADAMS) Limited in Hong Kong and OCUS Innovation Limited (HK, Ireland and UK), a former director of Therakind in England and Asia Medicine Regulatory Affairs (AMERA) Services Limited in Hong Kong. He is a former member of Pharmacy and Poisons Board, Hong Kong SAR Government. LW reports grants from the UK Cure Parkinson's Trust and UK NIHR. KKCM reports grants from CW Maplethorpe Fellowship, the European Union Horizon 2020, the UK National Institute of Health Research, the South Korea Ministry of Food and Drug Safety, the Hong Kong Research Grant Council and Hong Kong Innovation and Technology Commission, consultancy from IQVIA, AstraZeneca, outside of the submitted work. BC and FMCB declare that no competing interests exist.

**Abbreviations:** aIRR, adjusted incidence rate ratio; aOR, adjusted odds ratio; CCTC, case-case-time-control; CI, confidence interval; CPRD, Clinical Practice Research Datalink; HES, Hospital Episode Statistics; NSAIDs, non-steroidal anti-inflammatory drugs; ONS, Office for National Statistics; SCCS, self-controlled case series; SD, standard deviation; STROBE, Strengthening the Reporting of Observational Studies in Epidemiology.

## What did the researchers do and find?

- The findings suggest that gabapentinoids are often started during periods of already heightened vulnerability to drug poisoning, and that risk remains modestly elevated during treatment, and further elevated when opioids or benzodiazepines are co-prescribed.

- The risk of drug poisoning was more than doubled in the 90 days before starting gabapentinoids (aIRR = 2.09, 95% CI [1.98, 2.21]; $p < 0.001$), remained elevated in the first 28 days of treatment (aIRR = 1.81, 95% CI [1.66, 1.99]; $p < 0.001$) and was still raised during the remaining treatment period (aIRR = 1.11, 95% CI [1.05, 1.17]; $p < 0.001$) compared with non-treatment periods.

- Concurrent opioid/ benzodiazepine use further increased the risk, and the additional case-case-time-control analysis showed higher odds of gabapentinoid exposure before drug poisoning (aOR = 1.36, 95% CI [1.12,1.65]; $p = 0.002$), supporting the main findings.

## What do these findings mean?

- The findings suggest that gabapentinoids are often started during periods of already heightened vulnerability to drug poisoning, and that risk remains modestly elevated during treatment, and further elevated when opioids or benzodiazepines are co-prescribed.

- Clinicians should be aware of this increased vulnerability, monitor patients closely around treatment initiation, and avoid or minimise concurrent opioid or benzodiazepine use where possible to support safer prescribing.

- The SCCS approach cannot fully rule out the influence of unmeasured time-varying confounders, such as transient changes in socioeconomic status, major life events, or illicit drug use, although the negative control analysis did not indicate substantial residual confounding.

## Introduction

Gabapentinoids, including gabapentin and pregabalin, are structural analogues of gamma-aminobutyric acid that modulate neuronal excitability by binding to the alpha-2-delta subunit of voltage-gated calcium channels [1]. Initially approved for the treatment of seizures, gabapentinoids have since been widely prescribed for a variety of on- and off-label indications such as neuropathic pain, restless leg syndrome, anxiety disorders, insomnia, and bipolar disorder [2].

In recent years, there has been a substantial worldwide increase in gabapentinoid consumption [3]. It is also now the seventh most prescribed medication in the United States (US) [4]. This surge is partly attributed to their perceived safety profile and the quest for non-opioid analgesics [5–7]. However, the expanding use of gabapentinoids beyond their approved indications raises concerns about their potential for

misuse and adverse effects. Emerging evidence suggests that gabapentinoids possess abuse potential, especially among individuals with a history of illicit drug use disorders [7]. Reports have indicated that gabapentinoids can produce euphoria and enhance the effects of other central nervous system depressants, leading to increased risk of misuse and dependence [8]. A study by the Centers for Disease Control and Prevention found that gabapentin was detected in nearly one in 10 poisoning deaths in the United States between 2019 and 2020 [9]. A United Kingdom (UK) study has also shown that gabapentinoid-related poisoning fatalities have increased substantially in recent years, with 79% of them also involving the use of opioids [10]. Gabapentinoid-related poisoning can include both intentional self-poisoning, typically occurring in the context of self-harm, and non-intentional events such as accidental ingestions and misuse [11,12]. Some population-based and poison-centre studies report increased risks of suicidal behaviour and unintentional overdoses during gabapentinoid treatment in recent years [12,13]. Some studies suggest an elevated risk of overdose when gabapentinoids are used concomitantly with opioids or benzodiazepines, highlighting a potential synergistic effect [14–16]. However, these studies often focussed on specific populations, such as individuals with opioid use, limited to surgical patients only, and adopted a study design that may not fully account for confounders. Consequently, there is a gap in understanding the temporal relationship between gabapentinoid initiation and drug poisoning risk in the general population.

In this study, we hypothesised that initiation of gabapentinoid treatment is associated with an increased incidence of all-cause drug poisoning, and that this risk differs across predefined treatment windows. We also hypothesised that concurrent opioid or benzodiazepine use would modify this association, increasing the risk during overlapping treatment periods. To test these hypotheses, we planned and performed self-controlled case series (SCCS) analysis, comparing incidence rates of drug poisoning within individuals across non-treatment, pre-treatment, and post-initiation risk windows. This study design accounts for the diverse range of indications for gabapentinoids, which addresses the different underlying risks of drug poisoning associated with these indications, and accounting for all time-invariant confounders. The aim of this study is not to examine the underlying biological mechanisms linking gabapentinoid treatment to drug poisoning, but rather to evaluate their association in routine clinical practice.

## Methods

### Data sources

This study utilised data from the UK Clinical Practice Research Datalink (CPRD) Aurum, which was linked to the Hospital Episode Statistics (HES) and Office for National Statistics (ONS) databases from England. The database encompasses data from approximately 40 million patients across nearly 1,500 general practices [17], and it is representative of the general population of England for age, sex, and ethnicity [18]. Medical diagnoses and procedures are documented using the Read code and SNOMED-CT classification systems, while prescription information is captured through a drug dictionary derived from the British National Formulary [19]. The reliability of the data recorded in the CPRD has been demonstrated by prior research [20,21]. The profile of CPRD Aurum has been described in a published article [18]. The HES database contains hospital admission records of patients who have received care from National Health Services England hospitals [22]. Diagnoses in the HES are recorded using the International Classification of Diseases, 10th revision (ICD-10) classification [22]. The ONS database was used to accurately identify patients who died during follow-up and their cause of death.

### Study design

The main analysis adopted the SCCS [23,24] design to investigate the association between gabapentinoid treatment and drug poisoning. SCCS has previously been used to investigate the safety effects, including drug poisoning, of different medications in various health conditions [25–28]. SCCS includes patients who have both the outcome and treatment of interest within a predefined period [23,29]. Patients serve as their own controls [23]; hence, the major advantage of

this design is that it removes all time-invariant confounders, whether measured or unmeasured, which vary between individuals.

## Study participants

Individuals aged 18 years or above who received at least one prescription of gabapentinoids (S1 Table) and had their first HES record of all-cause drug poisoning dated during the study period (1st January, 2010, to 31st December, 2020) were identified (S2 Table). Individual observation periods commenced on the latest of 1st January, 2010, one year after the individual's CPRD registration date, or the 18th birthday and ended on the earliest of 31st December, 2020, the date of death, date the individual's registration at the practice ended, or diagnosis date of epilepsy or cancer. Patients with epilepsy or cancer occurring before the start of the observational period were excluded or censored on the date of diagnosis if occurring after, as they have different drug usage patterns and risk of drug poisoning [30,31]. Individuals who had gabapentinoid prescriptions one year before observation start were removed to account for the potential residual effect of previous treatments. Any individuals for whom the event occurred on the first day of gabapentinoid treatment periods or with missing information on year of birth or sex were also excluded. Fig 1 illustrates the selection of the study population.

## Exposures and outcomes

We identified all gabapentinoid prescriptions for each included individual. All gabapentinoid formulations and strengths were included in the analysis. We defined treatment periods as the time individuals were receiving gabapentinoids. We used the recorded prescription duration, quantity and daily doses prescribed to determine the duration of treatment. Gabapentinoid prescriptions that were less than or equal to 90 days apart were treated as a continuous treatment period.

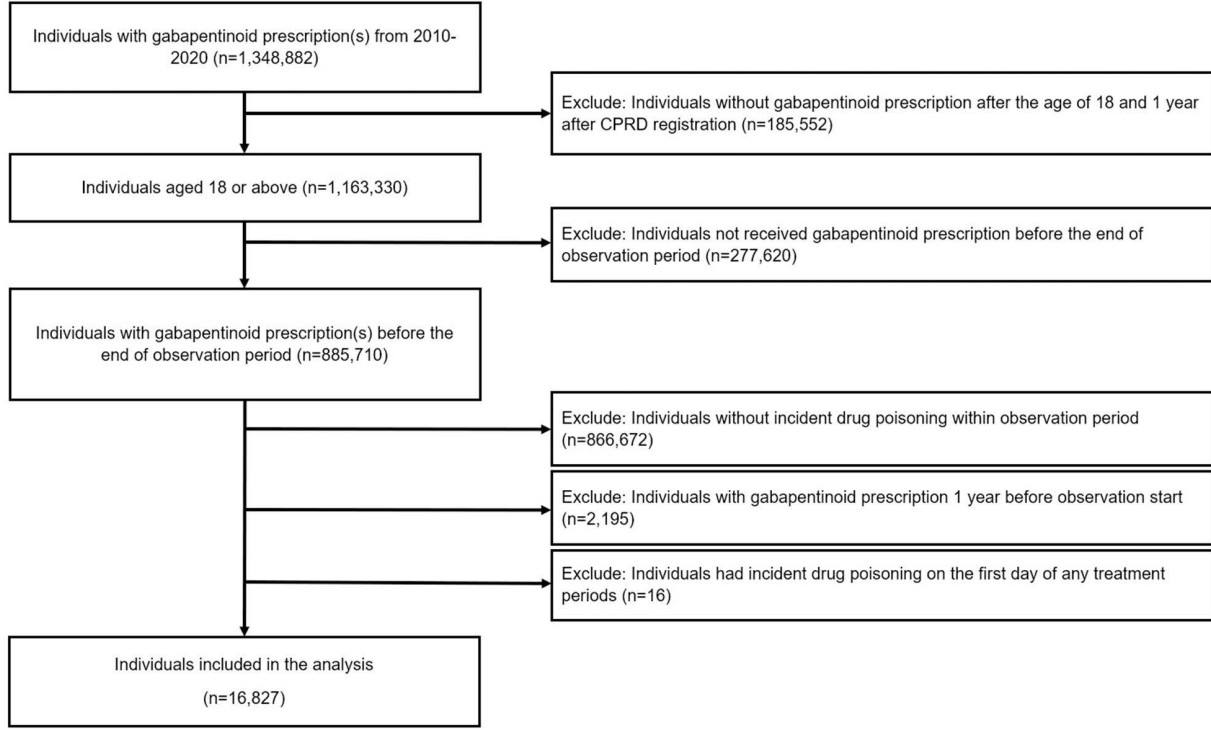

**Fig 1. Flowchart of patient identification.**

Gabapentinoid daily dose was calculated separately for gabapentin and pregabalin and converted into defined daily dose (DDD) (S2 Appendix), which is the assumed average maintenance dose per day for a drug used for its main indication in adults, developed by the World Health Organization [32].

We divided patient-time into six mutually exclusive risk windows: (1) 90 days before gabapentinoid treatment, (2) first 28 days of treatment periods, (3) 29–56 days of treatment periods, (4) 57–84 days of treatment periods, (5) remaining time of treatment periods, and (6) other non-treatment reference periods, where patient time does not belong to any of the previous risk windows (Fig 2). A 90-day period before treatment was added to account for the possibility that the episode of drug poisoning may affect the likelihood of gabapentinoid treatment, which in turn may introduce bias into the risk estimate during treatment [23].

All-cause drug poisoning diagnoses were defined as mental and behavioural disorders due to psychoactive substances (ICD-10, F11-F16, F18-F19), poisoning by drugs, medicaments and biological substances (T36-T50), accidental poisoning (X40-X44), intentional self-poisoning (X60-X64), assault by drugs (X85) and poisoning by drugs, medicaments and biological substances – undetermined intent (Y10-Y14) (S2 Table) [33–35]. The corresponding date of incident drug poisoning diagnosis from HES was identified as the event date. To avoid differential misclassification across drug poisoning intent categories, we adopted a composite definition using combinations of ICD-10 drug-poisoning diagnosis codes. This approach is consistent with previous epidemiological studies investigating risk factors associated with drug poisoning [34–37].

Interactions between gabapentinoid and opioid or benzodiazepine were also investigated, where included patients were required to receive both gabapentinoid and opioid or benzodiazepine during the observation period. Opioid or benzodiazepine risk windows were defined as three mutually exclusive windows, (A) 90 days before opioid/ benzodiazepine treatment, (B) opioid/ benzodiazepine treatment period, and (C) other non-treatment reference period, where patient time does not belong to any of the previous risk windows (S1 Appendix; S1 and S2 Figs). A total of 18 risk windows were defined to account for all possible risk window combinations between gabapentinoid and opioid/ benzodiazepine within the observation period and allow us to examine the risk throughout the treatment journey. To evaluate the difference of drug poisoning

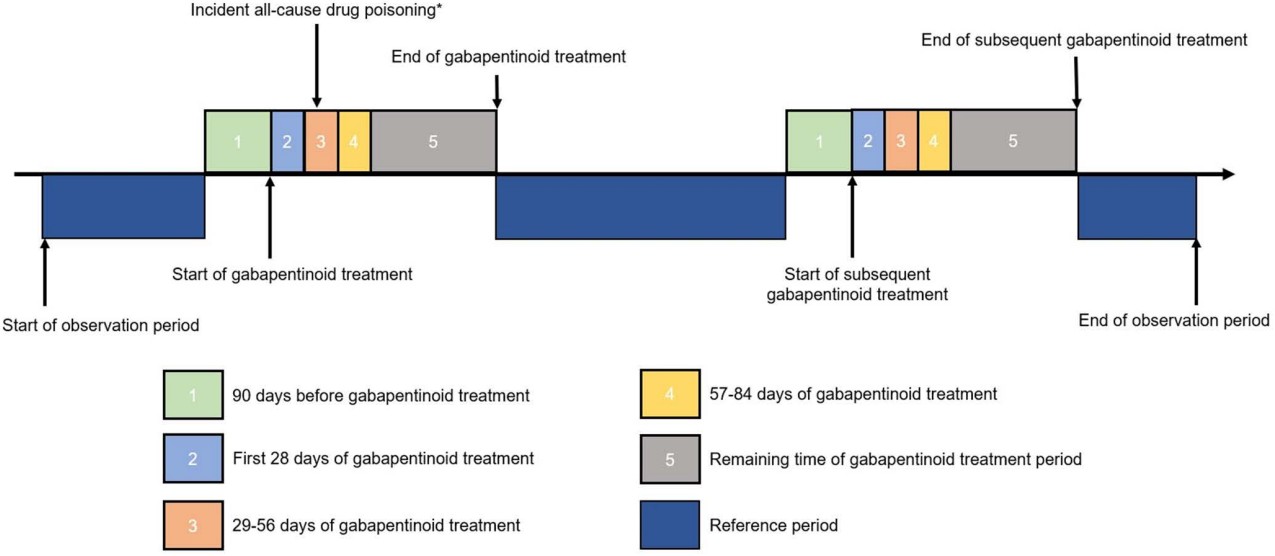

**Fig 2. Self-controlled case-series study design.** Illustration of the study design and timeline for a single hypothetical participant with an incident all-cause drug poisoning event. *Event can happen at any time throughout the observation period. GABA, gabapentinoids.

risk between gabapentin and pregabalin, a comparison analysis between gabapentin-only and pregabalin-only treatment periods in patients who took both medications was also conducted (S3 Fig).

## Statistical analysis

Crude incidence rates of drug poisoning in different risk periods were calculated. Adjusted incidence rate ratios (aIRRs) were estimated using conditional Poisson regression by comparing the incidence rate of events during treatment periods with reference periods, adjusted for age in 1-year bands, season (in three-month intervals) and concomitant opioids, anti-seizure medications, psychotropic medications and non-steroidal anti-inflammatory drugs (NSAIDs) (S3–S10 Tables) [38–40], which potentially affects gabapentinoid use and the risk of drug poisoning. Results were stratified by sex, age groups, ethnic groups, types of gabapentinoid, daily dose levels and different underlying comorbidities status (S2 Appendix). Ethnic groups were categorised into 5 groups which are Black, South Asian, White, Others and Missing. They are recorded in HES and also defined by SNOMED-CT code from CPRD Aurum. In secondary analyses, we used alternative outcome definitions to capture different types of drug poisoning, specifically accidental poisoning and intentional self-poisoning (S2 Appendix) [35]. Each outcome definition was analysed in a separate SCCS model using the same exposure definitions and adjustment strategy as in the primary analysis.

In the analyses for interaction between gabapentinoid and opioid/ benzodiazepine, the incidence rates for all-cause drug poisoning during different risk windows were compared to the incidence rate in the non-treatment reference period, without any exposure to gabapentinoid and opioid/ benzodiazepine. In the comparison study between gabapentin and pregabalin, the incidence rate of all-cause drug poisoning during the gabapentin-only treatment period was compared to the pregabalin-only treatment period.

## Sensitivity and negative control analyses

A series of pre-specified sensitivity analyses tested the validity and robustness of the main results: (a) spline-based SCCS analysis; (b) excluding patients who died within six months of the event [23]; (c) starting the observation period from the first neuropathic or chronic pain diagnosis (S4 Fig); (d) limiting the cohort to individuals with at least two gabapentinoid prescriptions; (e) adjusting the length of the pre-treatment period; (f) adjusting only for age and season; (g) extending treatment periods; (h) not combining gabapentinoid prescriptions if they were less than or equal to 90 days apart (an unplanned sensitivity analysis); (i) SCCS extension of event-dependent observation and exposure [41,42], which tested the key assumptions of the SCCS model [23,43]. Detailed information of all sensitivity analyses is provided in S3 Appendix. We also conducted a negative control analysis using food poisoning (S11 Table) as an outcome to identify any residual confounders.

An additional case-case-time-control (CCTC) analysis [44] was performed to validate the results from the SCCS study (S3 Appendix). The 30 days immediately preceding the incident all-cause drug poisoning event were designated as the hazard period and compared to four randomly selected 30-day reference periods occurring between 61 and 180 days prior to the event date (S5 Fig). Future cases were defined as individuals who experienced an incident drug poisoning event within 180–360 days of the current case and were matched by age, sex, and ethnicity. Conditional logistic regression was used to estimate the adjusted odds ratio (aOR) for exposure to gabapentinoids, at hazard period.

A two-sided significance level of 5% was used in all statistical analyses. SAS, version 9.4 and R Foundation for Statistical Computing version 4.2.0 were used for data analysis. This study is reported as per the Strengthening the Reporting of Observational Studies in Epidemiology (STROBE) guideline (S1 Checklist) [45]. Analyses were performed in accordance with the pre-registered study protocol approved by the Independent Scientific Advisory Committee of CPRD (S1 Protocol). A post hoc analysis in which gabapentinoid prescriptions issued 90 days or less apart were not combined was additionally conducted in response to reviewer comments.

### Ethics statement

Ethical approval was obtained from the Independent Scientific Advisory Committee of CPRD (protocol number: 23_002896). Informed consent was not required due to the use of de-identified data.

## Results

### Patient characteristics

The CPRD Aurum contained records of 1,348,882 patients who received at least one prescription for gabapentinoid between 1st January, 2010 and 31st December, 2020. 16,827 individuals met the inclusion criteria and were included in the SCCS analysis (Fig 1). Of the included cohort, 9,007 (53.5%) were female, the mean (standard deviation [SD]) age at the event was 46.91 (16.80) years, and the mean duration (SD) of the follow-up per individual was 8.18 (3.03) years (Tables 1 and S12). The median length of each gabapentinoid prescription was 28 days (Interquartile range 7–28 days) with a mean duration (SD) of gabapentinoid treatment 1.83 (2.27) years. 7,635 (45.4%) took gabapentin only and 5,842 (34.7%) took pregabalin only during the observation period. Before the incident drug poisoning event, 14,082 (83.7%) were diagnosed with neuropathic/chronic pain, 8,008 (47.6%) had a diagnosis of illicit drug use and 12,825 (76.2%) were diagnosed with some forms of mental health conditions (Table 1). 14,317 (85.1%) of the included patients had been prescribed opioids, neuropsychiatric medications or NSAIDs during the 6 months before the event. Antidepressants (n = 10,863, 64.6%) were the most prescribed medication, followed by opioids (n = 8,803, 52.3%), then gabapentinoids (n = 7,270, 43.2%). Antidepressants and opioids were also the most prescribed medications within the observation period.

### Association between gabapentinoids and all-cause drug poisoning

The overall incidence of all-cause drug poisoning in the 1,348,882 individuals was 5.45 per 1,000 patient-years during gabapentinoid treatment periods. In the 16,827 cases, the crude incidence of all-cause drug poisoning per 100 patient-years was 27.06 (95% confidence interval [CI] [25.73,28.39]) in the 90 days before treatment period, 27.12 (95% CI [24.79, 29.45]) in the first 28 days of treatment, 20.84 (95% CI [18.37, 23.32]) between 29 and 56 days of the treatment period, 17.98 (95% CI [15.53, 20.43]) during treatment days 57–84, 13.10 (95% CI [12.66, 13.53]) during the remaining time of treatment period and 10.67 (95% CI [10.47, 10.87]) in the non-treatment reference period (Table 2 and Fig 3). The risk of all-cause drug poisoning was more than doubled during the 90 days before gabapentinoid prescription (aIRR = 2.09, 95% CI [1.98, 2.21]; $p < 0.001$). The risk remains elevated by 80% during the first 28 days of treatment (aIRR = 1.81, 95% CI [1.66, 1.99]; $p < 0.001$) and decreased (aIRR = 1.46, 95% CI [1.29, 1.65]; $p < 0.001$) between 29 and 56 days of treatment period. The risk further decreased during treatment days 57–84 (aIRR = 1.27, 95% CI [1.10, 1.46]; $p < 0.001$) but remained elevated above reference level for the remaining time of treatment period (aIRR = 1.11, 95% CI [1.05, 1.17]; $p < 0.001$). The spline-based SCCS analysis demonstrates a consistent risk pattern with an increasing risk before the start of treatment, followed by a further decline after treatment initiation (S6 Fig). Stratified analyses focussing solely on gabapentin or pregabalin mirrored the main findings.

### Interaction with opioids or benzodiazepines

In the interaction analyses between gabapentinoid and opioid or benzodiazepine (Table 2 and Fig 4), concurrent use of opioid or benzodiazepine with gabapentinoid further increased the risk of all-cause drug poisoning in all risk windows of gabapentinoid treatment. In the first 28 days of treatment period, the aIRRs of concurrent use with opioids or benzodiazepines are 2.14 (95% CI [1.77, 2.58]; $p < 0.001$) and 3.95 (95% CI [3.07, 5.07]; $p < 0.001$), respectively. Concomitant use of opioids, benzodiazepines, and gabapentinoids is also associated with more than 3-fold increased risk of all-cause drug poisoning when compared to periods where individuals were not exposed to either of them (S13 Table). We found

**Table 1. Patient characteristics in relation to events. Values are numbers (percentages) unless stated otherwise.**

| Variables | Study population (*n* = 16,827) |
|---|---|
| **Mean age (SD) on event date (years)** | 46.91 (SD: 16.80) |
| **Mean follow-up time (SD) (years)** | 8.18 (SD: 3.03) |
| **Use of gabapentinoids during observation period** | |
| Prescribed with Gabapentin only | 7,635 (45.4%) |
| Prescribed with Pregabalin only | 5,842 (43.7%) |
| Prescribed with both Gabapentin and Pregabalin | 3,350 (19.9%) |
| **Comorbidities status before event** | |
| Neuropathic pain or chronic pain | 14,082 (83.7%) |
| Substance misuse | 8,008 (47.6%) |
| Bipolar and mania | 573 (3.4%) |
| Depression | 9,929 (59.0%) |
| Anxiety disorders | 8,469 (50.3%) |
| Schizophrenia | 323 (1.9%) |
| Other psychosis | 612 (3.6%) |
| Insomnia | 3,363 (20.0%) |
| Any of the above mental health conditions | 12,825 (76.2%) |
| Any of the above conditions | 16,260 (96.6%) |
| **Patients died within 6 months of event** | 448 (2.7%) |
| **Patients who received gabapentinoid treatment after event** | 13,064 (77.6%) |
| **Use of gabapentinoids 6 months before event** | |
| Gabapentinoids | 7,270 (43.2%) |
| Gabapentin | 3,041 (18.1%) |
| Pregabalin | 2,877 (17.1%) |
| **Use of other medications 6 months before event** | |
| Antiseizure medications | 835 (5.0%) |
| Opioids | 8,803 (52.3%) |
| Hypnotics and anxiolytics, except benzodiazepines | 2,916 (17.3%) |
| Benzodiazepines | 3,698 (22.0%) |
| Antidepressants | 10,863 (64.6%) |
| Antipsychotics | 2,473 (14.7%) |
| Lithium | 148 (0.9%) |
| NSAIDs | 4,618 (27.4%) |
| Any of the above | 14,317 (85.1%) |
| **Use of other medications during observation period** | |
| Antiseizure medications | 2,349 (14.0%) |
| Opioids | 14,978 (89.0%) |
| Hypnotics and anxiolytics, except benzodiazepines | 8,088 (48.1%) |
| Benzodiazepines | 9,207 (54.7%) |
| Antidepressants | 15,403 (91.5%) |
| Antipsychotics | 7,152 (42.5%) |
| Lithium | 380 (2.3%) |
| NSAIDs | 13,321 (79.2%) |
| Any of the above | 16,746 (99.5%) |

SD, standard deviation; NSAIDs, non-steroidal anti-inflammatory drugs.

no significant difference in the risk of all-cause drug poisoning between gabapentin and pregabalin when comparing gabapentin-only to pregabalin-only treatment periods (aIRR = 1.04, 95% CI [0.90, 1.21]; $p$ = 0.61) (Table 2).

**Subgroups, secondary and sensitivity analyses**

Most of the subgroup and secondary analyses showed a similar risk pattern (S14–S18 Tables and S7–S10 Figs). All sensitivity analyses findings were consistent with the main results (S19–S21 Tables and S11 Fig). The analysis excluding patients who died 6 months after event had no impact on the results. No significant association was found in the negative control analysis during all risk periods (S22 Table). Results from the CCTC analysis showed an increased odds (aOR = 1.36, 95% CI [1.12, 1.65]; $p$ = 0.002) under gabapentinoid treatment within the 30-day period before event (S23 Table).

## Discussion

In individuals who were exposed to gabapentinoids and had all-cause drug poisoning, the incidence was highest in the 90-day period before treatment, suggesting that the start of treatment tends to coincide with a period of increased risk of drug poisoning. Although the incidence of drug poisoning gradually declined after treatment initiation, the risk remained elevated throughout gabapentinoid treatment period and did not return to non-treatment reference level. These findings indicate that gabapentinoid use is associated with a higher risk of all-cause drug poisoning, especially at the initial phase of treatment.

Multiple factors may explain why gabapentinoid treatment initiation and increased risk of drug poisoning coincide. Given the wide range of gabapentinoid indications, treatment initiation may be due to concerns of a patient's worsening symptoms of conditions such as pain, anxiety, insomnia or other psychiatric disorders. These conditions are documented to be associated with increased risk of drug poisoning in previous studies [46–48]. Consistent with this hypothesis, 96% of the cohort were diagnosed with neuropathic/chronic pain, illicit drug use or psychiatric disorders and over 85% of the individuals were prescribed mood stabilisers, opioids, NSAIDs or other psychiatric medications within 6 months before incident event, highlighting the potential escalation of risk factors leading up to drug poisoning. The prescribing of gabapentinoids is likely to reflect a clinical response to worsening underlying conditions or an attempt to minimise the risk of drug poisoning by switching from alternative treatments. However, our data do not allow us to determine the precise clinical reasons for the heightened risk observed in the 90 days preceding gabapentinoid initiation, and further research is warranted. Moreover, the results in this study cannot be interpreted as gabapentinoid having an immediate effect on lowering the risk of drug poisoning since the included cohort still had an elevated risk of drug poisoning during the early phase of treatment and the risk did not return to unexposed reference level along the treatment journey.

This persistent elevated risk during gabapentinoid treatment is consistent with the findings from previous studies [13,15,16], despite differences in study designs and settings. The increased risk observed may reflect the exacerbating role of polydrug use in drug poisoning, suggesting that caution is warranted after initiating gabapentinoids. Gabapentinoids have also been documented to have reinforcing potential for individuals via euphoria and relaxation [49]. This phenomenon is further supported by our interaction analyses between gabapentinoids and opioids or/and benzodiazepines, demonstrating that concurrent use of these substances is associated with elevated risk of drug poisoning, with benzodiazepines showing a stronger synergistic effect. Our results also echo previous studies that highlighted the synergistic effect of gabapentinoids with opioids and benzodiazepines on drug poisoning risk [15,16].

This study employed a within-individual design investigating the risk of all-cause drug poisoning of gabapentinoids with the adjustment of age, season and concomitant medications. The within-individual design was adopted as gabapentinoid-treated and untreated patients can differ in important ways, especially with its wide range of indications. The increased risk before treatment has not been previously observed and may have been missed in a classic cohort study in which patients with either events or exposures before the commencement of the study are usually excluded. The use of a large database linked to hospital care provided sufficient statistical power to evaluate the association between

**Table 2. Results from the SCCS analysis, stratified by sex, types of gabapentinoids (mutually exclusive) and concomitant use with opioids or benzodiazepine.**

| | Number of Events | Patient-years | Crude incidence (per 100 patient-years) (95% CI) | aIRR* (95% CI) | P value |
|---|---|---|---|---|---|
| **Main Analysis (n = 16,827)** | | | | | |
| 90 days before treatment | 1,588 | 5,868.81 | 27.06 (25.73, 28.39) | 2.09 (1.98, 2.21) | <0.001 |
| First 28 days of treatment period | 520 | 1,917.38 | 27.12 (24.79, 29.45) | 1.81 (1.66, 1.99) | <0.001 |
| 29-56 days of treatment period | 273 | 1,309.71 | 20.84 (18.37, 23.32) | 1.46 (1.29, 1.65) | <0.001 |
| 57-84 days of treatment period | 207 | 1,151.44 | 17.98 (15.53, 20.43) | 1.27 (1.10, 1.46) | 0.001 |
| Remaining time of treatment period | 3,459 | 26,412.91 | 13.10 (12.66, 13.53) | 1.11 (1.05, 1.17) | <0.001 |
| Reference period | 10,780 | 101,052.50 | 10.67 (10.47, 10.87) | 1.00 (1.00, 1.00) | NA |
| **Stratified by sex** | | | | | |
| **Female (n = 9,007)** | | | | | |
| 90 days before treatment | 802 | 3,228.38 | 24.84 (23.12, 26.56) | 1.93 (1.79, 2.09) | <0.001 |
| First 28 days of treatment period | 281 | 1,050.93 | 26.74 (23.61, 29.86) | 1.82 (1.61, 2.06) | <0.001 |
| 29–56 days of treatment period | 133 | 715.73 | 18.58 (15.42, 21.74) | 1.32 (1.11, 1.58) | 0.002 |
| 57–84 days of treatment period | 92 | 629.02 | 14.63 (11.64, 17.61) | 1.05 (0.85, 1.29) | 0.67 |
| Remaining time of treatment period | 1,885 | 14,477.47 | 13.02 (12.43, 13.61) | 1.11 (1.03, 1.20) | 0.004 |
| Reference period | 5,814 | 54,144.31 | 10.74 (10.46, 11.01) | 1.00 (1.00, 1.00) | NA |
| **Male (n = 7,820)** | | | | | |
| 90 days before treatment | 786 | 2,640.44 | 29.77 (27.69, 31.85) | 2.28 (2.11, 2.47) | <0.001 |
| First 28 days of treatment period | 239 | 866.46 | 27.58 (24.09, 31.08) | 1.80 (1.57, 2.06) | <0.001 |
| 29–56 days of treatment period | 140 | 593.97 | 23.57 (19.67, 27.47) | 1.61 (1.36, 1.92) | <0.001 |
| 57–84 days of treatment period | 115 | 522.43 | 22.01 (17.99, 26.04) | 1.54 (1.27, 1.86) | <0.001 |
| Remaining time of treatment period | 1,574 | 11,935.45 | 13.19 (12.54, 13.84) | 1.10 (1.01, 1.19) | 0.03 |
| Reference period | 4,966 | 46,908.16 | 10.59 (10.29, 10.88) | 1.00 (1.00, 1.00) | NA |
| **Stratified by types of gabapentinoids (mutually exclusive)** | | | | | |
| **Gabapentin only (n = 7,635)** | | | | | |
| 90 days before treatment | 639 | 2,475.81 | 25.81 (23.81, 27.81) | 1.93 (1.77, 2.11) | <0.001 |
| First 28 days of treatment period | 225 | 805.43 | 27.94 (24.29, 31.59) | 1.81 (1.58, 2.08) | <0.001 |
| 29–56 days of treatment period | 103 | 502.42 | 20.50 (16.54, 24.46) | 1.40 (1.15, 1.71) | 0.001 |
| 57–84 days of treatment period | 69 | 427.95 | 16.12 (12.32, 19.93) | 1.12 (0.88, 1.43) | 0.36 |
| Remaining time of treatment period | 1,250 | 9,002.06 | 13.89 (13.12, 14.66) | 1.15 (1.05, 1.26) | 0.003 |
| Reference period | 5,349 | 49,830.33 | 10.73 (10.45, 11.02) | 1.00 (1.00, 1.00) | NA |
| **Pregabalin only (n = 5,842)** | | | | | |
| 90 days before treatment | 648 | 1,733.25 | 37.39 (34.51, 40.26) | 2.58 (2.36, 2.83) | <0.001 |
| First 28 days of treatment period | 187 | 571.66 | 32.71 (28.02, 37.40) | 1.85 (1.59, 2.16) | <0.001 |
| 29–56 days of treatment period | 101 | 427.72 | 23.61 (19.01, 28.22) | 1.41 (1.15, 1.73) | 0.001 |
| 57–84 days of treatment period | 82 | 389.01 | 21.08 (16.52, 25.64) | 1.28 (1.03, 1.61) | 0.03 |
| Remaining time of treatment period | 1,201 | 9,765.99 | 12.30 (11.60, 12.99) | 0.94 (0.85, 1.03) | 0.17 |
| Reference period | 3,623 | 32,329.57 | 11.21 (10.84, 11.57) | 1.00 (1.00, 1.00) | NA |
| **Used both gabapentin and pregabalin during observation period (n = 3,348)** | | | | | |
| Time exposed to gabapentin only | 429 | 3193.41 | 13.43 (12.16, 14.71) | 1.04 (0.90, 1.21) | 0.61 |
| Time exposed to pregabalin only (reference) | 673 | 5103.61 | 13.19 (12.19, 14.18) | 1.00 (1.00, 1.00) | NA |
| **Used both gabapentinoid and opioid during observation period (n = 8,224)** | | | | | |
| Opioid at reference and 90 days before gabapentinoid treatment | 284 | 1231.32 | 23.06 (20.38, 25.75) | 2.02 (1.79, 2.30) | <0.001 |

*(Continued)*

**Table 2.** (Continued)

| | Number of Events | Patient-years | Crude incidence (per 100 patient-years) (95% CI) | aIRR* (95% CI) | *P* value |
|---|---|---|---|---|---|
| Opioid at reference and first 28 days of gabapentinoid treatment | 98 | 425.66 | 23.02 (18.46, 27.58) | 1.83 (1.49, 2.24) | <0.001 |
| Opioid at reference and 29–56 days of gabapentinoid treatment | 43 | 278.43 | 15.44 (10.83, 20.06) | 1.24 (0.92, 1.69) | 0.16 |
| Opioid at reference and 57–84 days of gabapentinoid treatment | 34 | 240.15 | 14.16 (9.40, 18.92) | 1.15 (0.82, 1.61) | 0.43 |
| Opioid at reference and remaining time of gabapentinoid treatment | 535 | 4616.88 | 11.59 (10.61, 12.57) | 1.01 (0.90, 1.14) | 0.86 |
| Opioid treatment period and 90 days before gabapentinoid treatment | 270 | 933.81 | 28.91 (25.46, 32.36) | 2.36 (2.06, 2.71) | <0.001 |
| Opioid treatment period and first 28 days of gabapentinoid treatment | 127 | 412.97 | 30.75 (25.40, 36.10) | 2.14 (1.77, 2.58) | <0.001 |
| Opioid treatment period and 29–56 days of gabapentinoid treatment | 70 | 284.30 | 24.62 (18.85, 30.39) | 1.78 (1.39, 2.27) | <0.001 |
| Opioid treatment period and 57–84 days of gabapentinoid treatment | 51 | 251.78 | 20.26 (14.70, 25.82) | 1.50 (1.13, 1.99) | 0.005 |
| Opioid treatment period and remaining time of gabapentinoid treatment | 874 | 5830.86 | 14.99 (14.00, 15.98) | 1.37 (1.22, 1.53) | <0.001 |
| Opioid treatment period and gabapentinoid at reference | 1,046 | 8318.04 | 12.58 (11.81, 13.34) | 1.25 (1.14, 1.36) | <0.001 |
| Both gabapentinoid and opioid at reference period (reference) | 3,821 | 38931.98 | 9.81 (9.50, 10.13) | 1.00 (1.00, 1.00) | NA |
| **Used both gabapentinoid and benzodiazepine during observation period (*n*=7,263)** | | | | | |
| Benzodiazepine at reference and 90 days before gabapentinoid treatment | 356 | 1844.20 | 19.30 (17.30, 21.31) | 1.97 (1.76, 2.21) | <0.001 |
| Benzodiazepine at reference and first 28 days of gabapentinoid treatment | 113 | 616.00 | 18.34 (14.96, 21.73) | 1.75 (1.44, 2.11) | <0.001 |
| Benzodiazepine at reference and 29–56 days of gabapentinoid treatment | 70 | 430.52 | 16.26 (12.45, 20.07) | 1.57 (1.24, 2.00) | <0.001 |
| Benzodiazepine at reference and 57–84 days of gabapentinoid treatment | 55 | 376.99 | 14.59 (10.73, 18.44) | 1.41 (1.07, 1.84) | 0.01 |
| Benzodiazepine at reference and remaining time of gabapentinoid treatment | 954 | 8910.45 | 10.71 (10.03, 11.39) | 1.13 (1.02, 1.24) | 0.02 |
| Benzodiazepine treatment period and 90 days before gabapentinoid treatment | 148 | 403.27 | 36.70 (30.79, 42.61) | 4.23 (3.54, 5.07) | <0.001 |
| Benzodiazepine treatment period and first 28 days of gabapentinoid treatment | 70 | 171.03 | 40.93 (31.34, 50.52) | 3.95 (3.07, 5.07) | <0.001 |
| Benzodiazepine treatment period and 29–56 days of gabapentinoid treatment | 37 | 119.83 | 30.88 (20.93, 40.83) | 3.15 (2.25, 4.40) | <0.001 |
| Benzodiazepine treatment period and 57–84 days of gabapentinoid treatment | 19 | 106.90 | 17.77 (9.78, 25.77) | 1.83 (1.16, 2.90) | 0.01 |
| Benzodiazepine treatment period and remaining time of gabapentinoid treatment | 445 | 2703.69 | 16.46 (14.93, 17.99) | 2.27 (1.98, 2.62) | <0.001 |
| Benzodiazepine treatment period and gabapentinoid at reference | 751 | 3955.61 | 18.99 (17.63, 20.34) | 2.78 (2.51, 3.07) | <0.001 |
| Both gabapentinoid and benzodiazepine at reference period (reference) | 3,438 | 38475.07 | 8.94 (8.64, 9.23) | 1.00 (1.00, 1.00) | NA |

*All estimates are adjusted for age in 1-year age-band, seasonal effect, antiseizure medications, opioids, psychiatric medications and non-steroidal anti-inflammatory drugs. *P* values were obtained from two-sided Wald tests.

SCCS, self-controlled case series; *n*, number of individuals included in the analysis; aIRR, adjusted incidence rate ratio; CI, confidence interval; NA, not applicable.

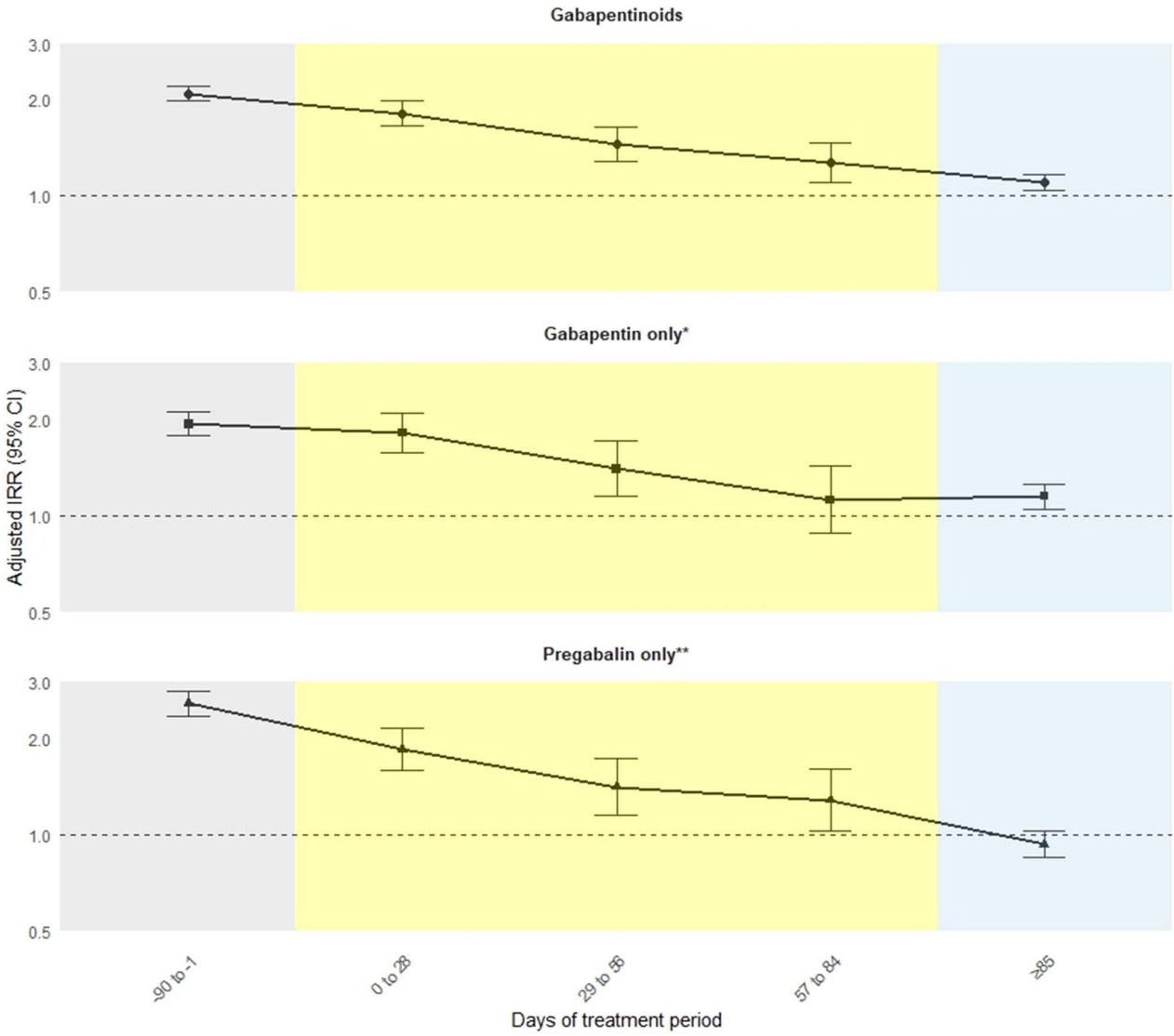

**Fig 3. Association between risk periods of gabapentinoid treatment and all-cause drug poisoning.** *Patients who took gabapentin only within the observation period were included in the analysis. **Patients who took pregabalin only within the observation period were included in the analysis. IRR, incidence rate ratio; CI, confidence interval.

gabapentinoid and drug poisoning in stratified risk windows and different subgroups of individuals. The within-individual design allows controlling for all time-invariant confounders by comparisons within individuals [24]. Important time-varying confounders which reflect the change in severity of underlying conditions were also adjusted in the regression models. The robustness of the SCCS results were also validated by the addition of CCTC analysis [44] and different SCCS extensions, including event-dependent observation [41], event-dependent exposure [42] and spline-based analyses [50]. The comprehensive range of subgroup and sensitivity analyses offers us a clearer picture of how gabapentinoid treatment can affect the risk of drug poisoning in patients with different underlying conditions, potentially informing healthcare professionals and patients about periods associated with a high risk of drug poisoning.

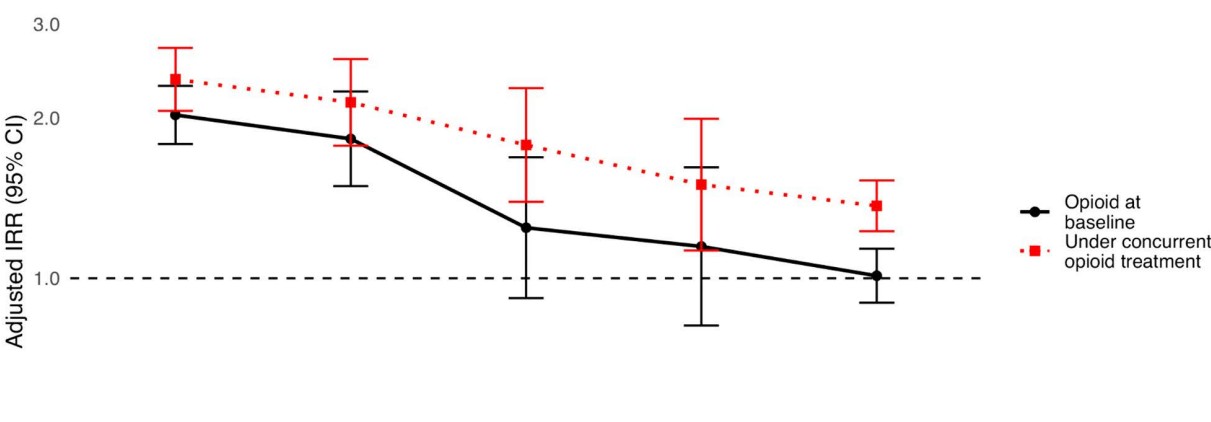

**Concomitant use with opioid**

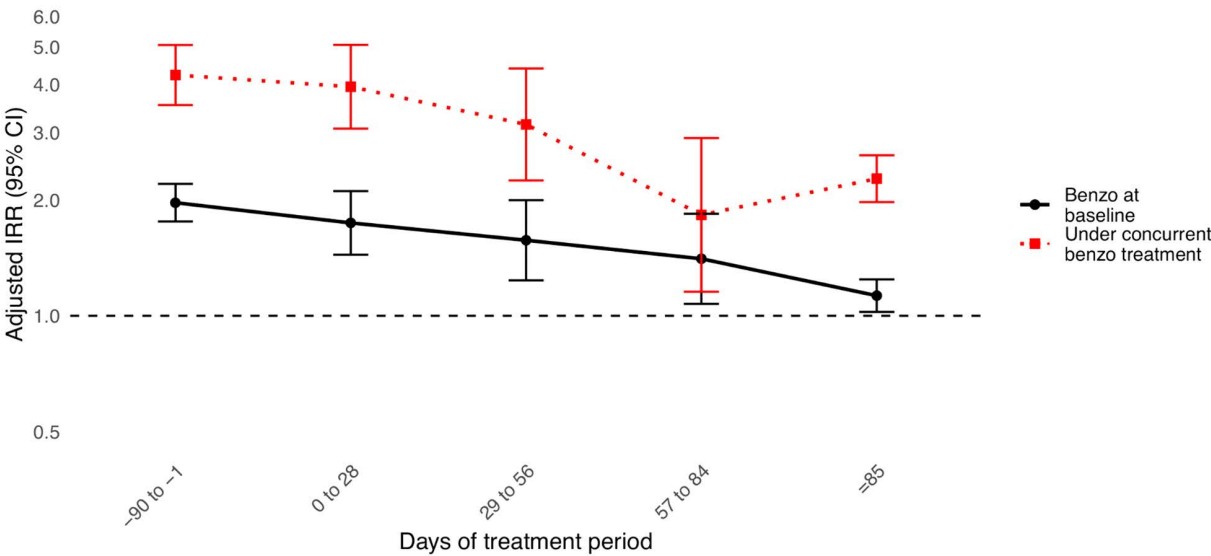

**Concomitant use with benzodiazepine**

**Fig 4. Association between gabapentinoid treatment and all-cause drug poisoning with concomitant use of opioids or benzodiazepines.**
Benzo, benzodiazepine; IRR, incidence rate ratio; CI, confidence interval.

Our study may have limitations. First, CPRD data does not include adherence information. Nonadherent patients may lead to exposure misclassification. However, the sensitivity analyses of extending treatment periods or analysing individuals with two or more prescriptions yielded results consistent with the main analysis. Second, the database only captured prescriptions from the general practitioner. Gabapentinoids prescribed in secondary or tertiary care or obtained through illegal means are not recorded. Third, some instances of drug poisoning may not result in secondary care attendance, potentially leading to an underestimation of the incidence rate. Fourth, in the two secondary analyses of accidental or intentional drug poisoning, the intention of drug poisoning may potentially be misclassified in clinical setting. Hence, we adopted a wider

definition of drug poisoning, including all incident outcomes disregarding their intention in our main analysis. Fifth, as pre-scribing records in CPRD are not linked to diagnostic codes, we were unable to identify the specific clinical indications for which gabapentinoids were prescribed at each treatment episode. Nevertheless, subgroup analyses according to underlying mental health comorbidities (S17 Table) yielded results that were broadly consistent with the main analysis. Sixth, our findings are based on routinely collected data from NHS primary and secondary care in England, and patterns of gabapentinoid prescribing, concomitant opioid or benzodiazepine use and diagnosis practice may differ in other healthcare systems. The observed risks may not be directly generalisable to countries with different model of care. Finally, similar to other observational studies, we cannot rule out the effect of unmeasured time-varying confounders such as transient socioeconomic status, life events and use of illicit drugs, although the negative control analysis result does not suggest issues with residual confounders. In addition, although the SCCS design controls for all time-invariant confounders and we adjusted for a range of concomitant medications as time-varying covariates, our study cannot fully account for time-varying changes in the severity of the underlying clinical conditions that prompted gabapentinoid initiation. Worsening pain, psychiatric symptoms, or other changes in health status may have contributed to the observed risk of drug poisoning around treatment initiation, and further studies are warranted to investigate the underlying clinical factors of this pattern.

The risk of drug poisoning increased in the period immediately preceding gabapentinoid initiation. Although the height-ened risk gradually declined as treatment progressed, the risk stayed consistently above non-treatment reference level, indicating that gabapentinoid therapy is associated with an increased risk of drug poisoning, particularly at the initial phase. Concomitant use with opioids or benzodiazepines further increased the risk of drug poisoning. The findings high-light the need for close monitoring of patients for drug poisoning throughout the treatment journey, and the importance of limiting concurrent use with opioid and benzodiazepines.

## Supporting information

**S1 Appendix. Comparison and interaction studies.**
(DOCX)

**S2 Appendix. Subgroups and secondary analyses.**
(DOCX)

**S3 Appendix. Sensitivity analyses and negative control analyses.**
(DOCX)

**S1 Table. Gabapentinoids included in the study.**
(DOCX)

**S2 Table. Diagnosis codes of all-cause drug poisoning.**
(DOCX)

**S3 Table. Antiseizure medications included in the study.**
(DOCX)

**S4 Table. Opioids included in the study.**
(DOCX)

**S5 Table. Hypnotics and Anxiolytics included in the study.**
(DOCX)

**S6 Table. Benzodiazepines included in the study.**
(DOCX)

**S7 Table. Antidepressants included in the study.**
(DOCX)

**S8 Table. Antipsychotics included in the study.**
(DOCX)

**S9 Table. Lithium included in the study.**
(DOCX)

**S10 Table. Non-steroidal anti-inflammatory drugs included in the study.**
(DOCX)

**S11 Table. Diagnosis codes of food poisoning.**
(DOCX)

**S12 Table. Patient characteristics.**
(DOCX)

**S13 Table. Results of concomitant use with gabapentinoids, opioids and benzodiazepines.**
(DOCX)

**S14 Table. Results of analyses stratified by age groups.**
(DOCX)

**S15 Table. Results of analyses stratified by ethnic groups.**
(DOCX)

**S16 Table. Results of analyses stratified by defined daily dose levels.**
(DOCX)

**S17 Table. Results of analyses stratified by comorbidities status.**
(DOCX)

**S18 Table. Results of accidental poisoning (ICD10, X40-X44) and intentional self-poisoning (ICD10, X60-X64).**
(DOCX)

**S19 Table. Results of sensitivity analyses.**
(DOCX)

**S20 Table. Sensitivity analysis results in the cohort with concomitant opioid or benzodiazepine use.**
(DOCX)

**S21 Table. Results of self-controlled case series extension analyses.**
(DOCX)

**S22 Table. Results of negative control analysis.**
(DOCX)

**S23 Table. Results of case-case-time-control analysis.**
(DOCX)

**S1 Fig. Interaction study design between gabapentinoids and opioids.** Illustration of the study design and timeline for a single hypothetical participant. *Event can happen at any time throughout the observation period. GABA, gabapentinoid.
(TIF)

**S2 Fig. Interaction study design between gabapentinoids and benzodiazepines.** Illustration of the study design and timeline for a single hypothetical participant. *Event can happen at any time throughout the observation period. GABA, gabapentinoid; Benzo, benzodiazepine.
(TIF)

**S3 Fig. Comparison study design between gabapentin-only and pregabalin-only treatment periods.** Illustration of the study design and timeline for a single hypothetical participant. *Event can happen at any time throughout the observation period.
(TIF)

**S4 Fig. Study design of observation started at neuropathic or chronic pain diagnosis.** Illustration of the study design and timeline for a single hypothetical participant. *Event can happen at any time throughout the observation period.
(TIF)

**S5 Fig. Case-case-time-control study design.** The case-case-time-control analysis incorporated two self-controlled analyses—a case crossover analysis and a control crossover analysis consisting of future cases to address confounding by indication and potential protopathic bias. OR, odd ratio.
(TIF)

**S6 Fig. Results from the spline-based self-controlled case series analysis.** The dotted lines represent the range of 95% confidence intervals.
(TIF)

**S7 Fig. Association between risk periods of gabapentinoid treatment and all-cause drug poisoning, stratified by sex.** IRR, incidence rate ratio; CI, confidence interval.
(TIF)

**S8 Fig. Association between risk periods of gabapentinoid treatment and all-cause drug poisoning, stratified by age groups.** IRR, incidence rate ratio; CI, confidence interval.
(TIF)

**S9 Fig. Results from the spline-based self-controlled case series analysis of accidental poisoning only.** The dotted lines represent the range of 95% confidence intervals.
(TIF)

**S10 Fig. Results from the spline-based self-controlled case series analysis of intentional self-poisoning only.** The dotted lines represent the range of 95% confidence intervals.
(TIF)

**S11 Fig. Sensitivity analysis on treatment periods by adding 28, 56 and 84 days after the end of a treatment period.** IRR, incidence rate ratio; CI, confidence interval.
(TIF)

**S12 Fig. Histogram of age at the incident all-cause drug poisoning event.**
(TIF)

**S1 Protocol. The risk of adverse psychiatric and somatic outcomes with gabapentinoid use: protocol of a UK population-based study using electronic health records.**
(DOCX)

**S1 Checklist. STROBE Statement—checklist of items that should be included in reports of observational studies.** This checklist is reproduced from the STROBE Statement (Strengthening the Reporting of Observational Studies in

Epidemiology) and is licensed under the Creative Commons Attribution 4.0 International (CC BY 4.0). von Elm E, Altman DG, Egger M, Pocock SJ, Gøtzsche PC, et al. (2007) The Strengthening the Reporting of Observational Studies in Epidemiology (STROBE) Statement: Guidelines for Reporting Observational Studies. PLOS Medicine 4(10): e296. https://doi.org/10.1371/journal.pmed.0040296.
(DOCX)

## Author contributions

**Conceptualisation:** Andrew S. C. Yuen, Joseph F. Hayes, Kenneth K. C. Man.

**Data curation:** Andrew S. C. Yuen, Kenneth K. C. Man.

**Formal analysis:** Andrew S. C. Yuen, Boqing Chen.

**Funding acquisition:** Andrew S. C. Yuen, Kenneth K. C. Man.

**Investigation:** Andrew S. C. Yuen, Boqing Chen, Adrienne Y. L. Chan, Joseph F. Hayes, David P. J. Osborn, Frank M. C. Besag, Wallis C. Y. Lau, Ian C. K. Wong, Li Wei, Kenneth K. C. Man.

**Methodology:** Andrew S. C. Yuen, Adrienne Y. L. Chan, Kenneth K. C. Man.

**Project administration:** Kenneth K. C. Man.

**Resources:** Andrew S. C. Yuen, Kenneth K. C. Man.

**Supervision:** Li Wei, Kenneth K. C. Man.

**Validation:** Andrew S. C. Yuen, Boqing Chen, Kenneth K. C. Man.

**Visualisation:** Andrew S. C. Yuen.

**Writing – original draft:** Andrew S. C. Yuen.

**Writing – review & editing:** Andrew S. C. Yuen, Boqing Chen, Adrienne Y. L. Chan, Joseph F. Hayes, David P. J. Osborn, Frank M. C. Besag, Wallis C. Y. Lau, Ian C. K. Wong, Li Wei, Kenneth K. C. Man.

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
