## [Editor Report · Decision Letter 0]

3 Sep 2025

Dear Dr Man,

Thank you for submitting your manuscript entitled "Association between gabapentinoid treatment, concurrent use with opioid or benzodiazepine and the risk of drug poisoning" for consideration by PLOS Medicine.

Your manuscript has now been evaluated by the PLOS Medicine editorial staff as well as by an academic editor with relevant expertise and I am writing to let you know that we would like to send your submission out for external peer review.

For clinical studies, please upload a copy of your trial study protocol as a supporting information file. The study protocol should be the version submitted for approval to the institutional review board or ethics committee, should include any amendments to the study protocol, as well as the date of their approval by the institutional review or ethics committee. Please also detail any deviations from the study protocol in the Methods section of your manuscript. The editors will consider the protocol and study conduct prior to a final decision for external review.

Please re-submit your manuscript within two working days, i.e. by Sep 05 2025 11:59PM.

Kind regards,

Heather Van Epps, PhD

Consulting Editor

PLOS Medicine

---

## [Decision Letter · Decision Letter 1]

21 Nov 2025

Dear Dr Man,

Many thanks for submitting your manuscript "Association between gabapentinoid treatment, concurrent use with opioid or benzodiazepine and the risk of drug poisoning" (PMEDICINE-D-25-03057R1) to PLOS Medicine. Please accept my apologies for the unusual delay in providing you with a decision, due to a busy time for the editorial team and some difficulties recruiting enough reviewers. The paper has been reviewed by subject experts and a statistician; their comments are included below and can also be accessed here: [LINK]

As you will see, two of the reviewers are very positive, whereas the remaining reviewer has several serious concerns. After discussing the paper with the editorial team and an academic editor with relevant expertise, I'm pleased to invite you to revise the paper in response to the reviewers' comments. We plan to send the revised paper to some or all of the original reviewers, and we may also recruit an additional reviewer if we deem this necessary. At this stage, we cannot provide any guarantees regarding publication.

In addition to these revisions, we also have several editorial requests, which you can find at the bottom of this email. Furthermore, you may need to complete some formatting changes, which you will receive in a follow up email. A member of our team will be in touch with a set of requests shortly. If you do not receive a separate email within a few days, please assume that checks have been completed, and no additional changes are required.

We ask that you submit your revision by Dec 12 2025 11:59PM. However, if this deadline is not feasible, please contact me by email, and we can discuss a suitable alternative.

Don't hesitate to contact me directly with any questions (sbruijn@plos.org).

Best regards,

Suzanne

Suzanne De Bruijn, PhD

Associate Editor

PLOS Medicine

sbruijn@plos.org

Comments from the reviewers:

Reviewer #1: Thanks for the opportunity to read your manuscript. My role is statistical reviewer, so I have focused on the design, data, and analysis that are presented. I have put general comments first, followed by questions relevant to a specific section of the manuscript (with a page/line reference).

This manuscript estimates the association of treatment with gabapentinoids with drug poisoning, and whether there is an interaction between gabapentinoids and other drug classes. Data is from the UK CPRD, which links general practice data with hospital, prescription and mortality data. The self-controlled case series design was used to estimate the association between exposure and drug poisoning. Participants were selected who had a hospital record of drug poisoning between 2010 and 2020 and excluded if they had epilepsy or cancer, or prescribed gabapentinoid in the year before the start of the temporal window or had an event on the first day of the exposure or missing age or sex data. Exposure to gabapentinoid was derived from prescription duration, based on daily dose and quantity supplied. Gaps of coverage less than 90 days were considered contiguous. The period 90 days before gabapentinoid treatment was used as the 'control' portion of the case series, and several temporal windows up to 84 days after initiating treatment. Exposure to opioids and benzos was included as an exposure - to estimate the interaction between gabapentinoid an 18 level variable that combines the 3 opioid/benzo exposure periods and the 6 gabapentinoid periods was created. Poisson regression was used for the main analysis, to estimate incidence rates in the 'treatment' period compared to the control periods. This included an adjustment for age, and concomitant medications, and results presented stratified by key personal variables and drug information. Several sensitivity analyses were considered, including a spline-based approach to differences across the study temporal window, and a negative control analysis with food poisoning as an outcome. The rationale for all of these is explained well. I agree with the authors that the results of the sensitivity analyses are in line with the main results, except for the negative control which does not seem to be associated with gabapentinoid exposure. The limitations of the study are articulated well (e.g. unmeasured time-varying confounders).

There was extensive supplementary material included - this was a great help in reviewing the paper, thank you. I enjoyed reading this manuscript - there were only a few small clarifications I think the manuscript needs.

As a non-expert in gabapentinoid treatment, I was unclear about the potential for non-intentional poisoning, there is some useful context about misuse, but an additional sentence or two that summarises non-intentional and intentional poisoning with gabapentinoid would be helpful.

P7, L137. Were there any drug poisoning episodes identified from mortality data not recorded in the HES?

P7, L141. Were participants included if the drug poisoning occurred before they were 18?

P8, L155. What is the typical prescription period for gabapentinoid in the UK?

P10, L196. Stratifying by different types of outcomes technically means you have defined additional outcomes, rather than doing a sub-group analysis. This might be better explained these terms.

P26, Table 1. A very minor point, but 1 decimal place for the percentages is probably sufficient.

Reviewer #2: Review for PMEDICINE-D-25-03057R1

This study attempts to examine the association between gabapentinoid treatment, concurrent use with opioids or benzodiazepines, and the risk of drug poisoning. The study used the UK Clinical Practice Research Datalink database between January 1, 2010, and December 31, 2020, and identified 16,827 patients who had a diagnosis of all-cause drug poisoning. The authors used a self-controlled case series (SCCS), a within-individual study design, to test the associations.

Major concerns

* Introduction: The rationale for studying all-cause drug poisoning is not well justified. The study included patients with drug poisonings, "defined as mental and behavioral disorders due to psychoactive substances (ICD-10, F11-F16, F18-F19), poisoning by drugs, medicaments and biological substances (T36-T50), accidental poisoning (X40-X44), intentional self poisoning (X60-X64), assault by drugs (X85) and poisoning by drugs, medicaments and biological substances undetermined intent (Y10-Y14) (Supplementary Table 2).[30-32]" The rationale for studying this wide range of causes for drug poisoning is not supported by potential mechanisms. Nor does it have evidence to support the study of combining all causes of drug poisoning, each of which likely differs in underlying biological and clinical mechanisms.

* While the SCCS design controls for potential confounders that did not change over time, it does not adjust for confounders that change over time, such as the presence of a disease indication for treatment or disease severity. The study findings are subject to confounding by indication and confounding by disease severity because the reference is a no-treatment period, a duration during which an individual has no disease diagnosis or has the diagnosis but the disease condition is not clinically severe enough to require treatment. The use of a no-treatment period likely makes the reference group artificially superior (low risk of outcomes) to any studied treatment periods, leading to a higher risk of all-cause drug poisoning in the treatment periods vs the non-treatment period.

* Conclusion (Page 14, lines 299-301): "…the incidence was highest in the 90-day period before treatment, suggesting that the start of treatment tends to coincide with a period of increased risk of drug poisoning." This conclusion is confusing and lacks clinical justification. Clinically, gabapentinoids are recommended (at least in the US) as an alternative pain treatment to opioids or as a co-use treatment with opioids to avert or reduce the risk of opioid use disorder or overdose. Is it possible that gabapentinoids were initiated as alternative treatment to further minimize drug poisoning after being diagnosed (that's why the risk was highest in the 90 days before the treatment initiation)?

* Conclusion (page 15, lines 304-305): "These findings indicate that gabapentinoid use is associated with a higher risk of all-cause drug poisoning, especially at the initial phase of treatment". In some clinical cases, gabapentinoids are prescribed alone or in combination with opioids when doctors suspect patients of having symptoms or behaviors of opioid misuse, before the diagnosis of opioid poisoning is formally made. In other words, doctors may selectively prescribe gabapentinoid to individuals for whom (or individual periods during which) symptoms of opioid-related or drug poisoning due to other causes have been present but not yet diagnosed. The big question is to what extent the observed elevated risks during the gabapentinoid treatment period are due to doctors' selective prescribing behaviors (i.e., channeling bias) or due to the treatment itself, which could not be teased out by the study design and data.

* Study participants: It is unclear what disease condition was treated by gabapentinoid treatment among the study sample. Could an individual be prescribed gabapentin for only one condition or different conditions over the study period? The potential disease condition triggering gabapentinoid prescribing was not accounted for in the analysis, leading to potential residual confounding.

* Exposure (page 8, lines 161-165): The rationale for dividing person-time into 6 mutually risk windows is not clinically justified.

* Exposure (page 8, line 155): "Gabapentinoid prescriptions that were less than or equal to 90 days apart were treated as a continuous treatment period." There is no justification for the use of 90 days as the cut-off point for the grace period. A 90-day gap for some conditions (neuropathic pain) could be clinically considered as treatment discontinuation.

Other concerns

* Of the identified 16,827 patients with all-cause drug poisonings, what is the distribution by types of drug poisoning?

Reviewer #3: Thank you for the opportunity to review this important paper addressing a priority challenge: increasing rates of use of gabapentin. This paper is well written and has robust sensitivity analyses etc. I had a few comments that I hope you can address: 1) While the authors mention the potential impact of time varying confounders, they do not address clinical time varying confounders that may precipitate a prescription for gaba and potentially put a patient at higher risk for poisoning. Minimally I hope the authors can talk more about this potential in their limitations or ideally provide some more of a breakdown of clinical characteristics in the pre vs post-GABA Rx period (not pre or post event period). 2) If the authors can comment specifically about any potential issues with generalizability of study findings beyond the context of the NHS system that would also be instructive for readers. Overall, a well written paper.

---

* Please upload any figures associated with your paper as individual TIF or EPS files with 300dpi resolution at resubmission; please read our figure guidelines for more information on our requirements: http://journals.plos.org/plosmedicine/s/figures. While revising your submission, we strongly recommend that you use PLOS's NAAS tool (https://ngplosjournals.pagemajik.ai/artanalysis) to test your figure files. NAAS can convert your figure files to the TIFF file type and meet basic requirements (such as print size, resolution), or provide you with a report on issues that do not meet our requirements and that NAAS cannot fix.

After uploading your figures to PLOS's NAAS tool - https://ngplosjournals.pagemajik.ai/artanalysis, NAAS will process the files provided and display the results in the "Uploaded Files" section of the page as the processing is complete.

If the uploaded figures meet our requirements (or NAAS is able to fix the files to meet our requirements), the figure will be marked as "fixed" above. If NAAS is unable to fix the files, a red "failed" label will appear above.

When NAAS has confirmed that the figure files meet our requirements, please download the file via the download option, and include these NAAS processed figure files when submitting your revised manuscript.

* We note that you state in the manuscript that you obtained ethical approval. Please amend your ethics statement in the metadata in Editorial Manager, to include the same statement there.

SUPPLEMENTARY MATERIAL

REFERENCES

OBSERVATIONAL STUDIES

* Abstract: Please include the study design, population and setting, number of participants, years during which the study took place (enrollment and follow up), length of follow up, and main outcome measures.

* Thank you for reporting your study according to the STROBE guidelines, and including the checklist. Please also add the following statement, or similar, to the Methods: "This study is reported as per the Strengthening the Reporting of Observational Studies in Epidemiology (STROBE) guideline (S1 Checklist)." When completing the checklist, please use section and paragraph numbers, rather than page numbers.

* For all observational studies, in the manuscript text, please indicate: (1) the specific hypotheses you intended to test, (2) the analytical methods by which you planned to test them, (3) the analyses you actually performed, and (4) when reported analyses differ from those that were planned, transparent explanations for differences that affect the reliability of the study's results. If a reported analysis was performed based on an interesting but unanticipated pattern in the data, please be clear that the analysis was data driven.

* Please state in the Methods section whether the study had a prospective protocol or analysis plan. If a prospective analysis plan (from your funding proposal, IRB or other ethics committee submission, study protocol, or other planning document written before analyzing the data) was used in designing the study, please include the relevant document(s) with your revised manuscript as a Supporting Information file to be published alongside your study and cite it in the Methods section. A legend for this file should be included at the end of your manuscript. If no such document exists, please make sure that the Methods section transparently describes when analyses were planned, and when/why any data-driven changes to analyses took place. Changes in the analysis, including those made in response to peer review comments, should be identified as such in the Methods section of the paper, with rationale.

---

## [Decision Letter · Decision Letter 2]

4 Mar 2026

Dear Dr. Man,

Thank you very much for re-submitting your manuscript "Association between gabapentinoid treatment, concurrent use with opioid or benzodiazepine and the risk of drug poisoning" (PMEDICINE-D-25-03057R2) for review by PLOS Medicine.

I have discussed the paper with my colleagues and the academic editor and it was also seen again by 3 reviewers. I am pleased to say that provided the remaining editorial and production issues are dealt with we are planning to accept the paper for publication in the journal.

You will see that reviewer #2 still has concerns. We discussed this with the academic editor, and would like you to provide a rebuttal to these comments, and include some additional text in the manuscript where necessary to further highlight the limitations of the study.

The remaining editorial issues that need to be addressed are listed at the end of this email. Any accompanying reviewer attachments can be seen via the link below. Please take these into account before resubmitting your manuscript:

[LINK]

We look forward to receiving the revised manuscript by Mar 11 2026 11:59PM.

Sincerely,

Suzanne De Bruijn, PhD

Associate Editor

PLOS Medicine

plosmedicine.org

Requests from Editors:

GENERAL

* Please confirm that your title complies with PLOS Medicine's style. Your title must be nondeclarative and not a question. It should begin with main concept if possible. "Effect of" should be used only if causality can be inferred, i.e., for an RCT. Please place the study design ("A randomized controlled trial," "A retrospective study," "A modelling study," etc.) in the subtitle (ie, after a colon).

* Title: we suggest to change the title to: “Association between gabapentinoid treatment, concurrent use with opioid or benzodiazepine and the risk of drug poisoning – a self-controlled case series study”.

* Please ensure that all abbreviations are defined at first use throughout the text.

* Please confirm that all numbers presented in the abstract are present and identical to numbers presented in the main manuscript text.

* Please review your text for claims of novelty or primacy (e.g. 'for the first time') and remove this language. In addition, please check that any use of statistical terms (such as trend or significant) are supported by the data, and if not please remove them.

* Please remove the 'conclusions' subheading from the discussion. Please also remove any other subheadings from the discussion.

* Statistical reporting: Thank you for formatting your manuscript to fit with PLOS Medicine guidelines. However, the tables have not been adjusted. Please revise the tables, and check the whole manuscript.

- Please report statistical information as follows to improve clarity for the reader ""22% (95% CI [13,28]; p</=)"".

- Please separate upper and lower bounds with commas instead of hyphens as the latter can be confused with reporting of negative values.

- Please repeat statistical definitions (HR, CI etc.) for each set of parentheses.

DATA AVAILABILITY STATEMENT

* DAS: Thank you for providing the code. However, I can’t find the page, could you please confirm the URL? Furthermore, We strongly recommend that all code be deposited in a permanent, public repository that issues citable digital object identifiers (DOI) or other persistent identifiers, such as Zenodo.

*Can you verify that the HES and ONS data are also obtained from CPRD? If not, please add how these datasets can be accessed in the data availability statement.

ABSTRACT

* In the abstract, please include the important dependent variables that are adjusted for in the analyses.

* First sentence abstract-background: please check whether this statement is correct.

*line 71: typo in Gabapentinoid, please correct.

AUTHOR SUMMARY

* In the author summary, please revise formatting and ensure you use bullet points.

* "why was this study done": consider adding a few words what gabapentinoids are used for.

*“What did the researchers do and find: first point”: please rewrite this in laymen's terms. Consider adding a description of the data used, rather than (or in addition to) the names of the databases. Also spell out the SCCS acronym, and consider adding a sentence of explanation on this study-type.

OBSERVATIONAL, COHORT, CROSS-SECTIONAL, AND CASE CONTROL STUDIES

* Did your study have a prospective protocol or analysis plan? Please state this (either way) early in the Methods section.

FIGURES AND TABLES:

* When a p value is given, please specify the statistical test used to determine it in the legend.

* Please consider if moving some of the figures shared in the supplementary information into the main text would aid the reader. Specifically, we would suggest moving figure S1 to the main text.

Comments from Academic Editor:

The authors have responded well to the comments. Reviewer 2 is concerns about possible biological mechanisms, which is not an aim of the study. They are also concerns about time-varying confounders, which is an inherent limitation to the design used, and the authors have used reasonable adjustments to address this. the use of negative controls is a further strength of the paper.

Comments from Reviewers:

Reviewer #1:

Thanks for the revised manuscript and responses to my original review. The revised manuscript and responses to my questions have resolved my original questions.

I wasn't clear in my original review about the finding for the negative control - I absolutely agree with the authors that the results for the sensitivity analysis strengthens the manuscript.

I am comfortable with the authors choice for a 90-day gap - this is time-period is commonly selected in pharmacoepidemiology studies of other medicines, and the justification provided in response (supported by the results of the sensitivity analysis) is sound.

Reviewer #2: Review for PMEDICINE-D-25-03057R2

While the authors have attempted to address the questions, major concerns remain as follows:

* Mechanisms between gabapentinoid treatment and all-cause drug poisoning remain unclear, which is also pointed out by the authors' response-- "the actual biological mechanism is yet to be understood". The authors provided new data that among the identified 16,827 patients with all-cause drug poisonings in the study, most of the patients had multiple poisoning diagnoses on the same day. This raises another big question: how likely is it that the use of gabapentin treatment could be so detrimental, leading to multiple types of all-cause drug poisoning being diagnosed for the same person at the same time? The underlying biological or clinical mechanisms are unlikely to be plausible. The rationale for studying all-cause drug poisoning remains not justified.

* The conclusion—"The results suggest that gabapentinod is associated with an increased risk of drug poisoning" — remains problematic and overstated. The limitations of data and design cannot tease out whether the occurrence of all-cause drug poisoning is mainly because of gabapentinoid treatment or is due to the underlying serious conditions (e.g., depression), which pain condition (potentially treated by gabapentin) is also prevalently co-occurred with. In other words, gabapentinoid may be a "good" drug and used to treat co-morbidities or symptoms of major conditions that cause all-cause drug poisoning. The points are also mentioned in the response by authors, "The prescribing of gabapentinoids is likely to reflect a clinical response to worsening underlying conditions or an attempt to minimise the risk of drug poisoning by switching from alternative treatments. However, our data do not allow us to determine the precise clinical reasons for the heightened risk observed in the 90 days preceding gabapentinoid initiation, and further research is warranted." (P.18, Line 383 to 388).

* Study participants remain concerning because CPRD data are not linked to diagnostic codes (P.20 to 21, lines 433 to 435)." The current selected study participants with gabapentinoid consist of a mixed group of patients who received the drug for treating various clinical conditions, complicating the study findings and interpretations.

* The limitation that CPRD data are not linked to diagnostic codes likely creates potential disease indication for treatment or disease severity for the findings, a major concern that has been brought up. In this response, the authors argued that "In the statistical analysis model, we explicitly adjusted for time-varying proxies of disease severity and clinical instability, especially concomitant prescriptions for opioids, antiseizure medications, hypnotics and anxiolytics, benzodiazepines, antidepressants, antipsychotics, lithium and NSAIDs." The statement is not supported by any references. What does clinical instability mean? In clinical settings, the use of these medications is rarely considered to be a valid indicator of clinical severity and clinical instability of the underlying treated conditions. The assertion is overstated.

Reviewer #3: Thank you for thoroughly addressing all the reviewer comments and offering additional analyses and insights to strengthen confidence in your work.

[LINK]

---

## [Editor Report · Decision Letter 3]

18 Mar 2026

Dear Dr. Man,

Thank you very much for re-submitting your manuscript "Association between gabapentinoid treatment, concurrent use with opioid or benzodiazepine and the risk of drug poisoning – a self-controlled case series study " (PMEDICINE-D-25-03057R3) for review by PLOS Medicine.

Thank you for addressing our comments, and providing a response to the remaining reviewer comments. Before we can accept, we would like you to address a final few points.

1) Thank you for stating that your GitHub page has been linked to Zenodo. Please also include the Zenodo URL in the Data Availability statement.

2) Thank you for making the requested changes to the author summary. However, the sentence for the first bullet point is now quite long. Please split this into 2 sentences for readability. We suggest:

“Use of gabapentinoids, which are indicated for neuropathic pain, epilepsy, and generalised anxiety disorder, and are also used off label for fibromyalgia, sleep disorders, and other chronic pain conditions, has increased substantially over the past decade. However, there are growing concerns that they may contribute to drug poisoning and overdose, particularly when combined with opioids or benzodiazepines.”

Thank you for providing a thoughtful response to the remaining reviewer comments. However, whereas the rebuttal was extensive, we noted that little of this made it into the MS. Please add something of this rebuttal to the paper

3) in relating to the first point, maybe clarify further that the study is not aimed at outlining biological mechanisms.

4) in regarding to the second point, could you add some additional text to the limitation section?

The third and fourth point do not need any further revision to the text.

We expect to receive your revised manuscript within 1 week. Please email me (sbruijn@plos.org) if you have any questions or concerns.

We look forward to receiving the revised manuscript by Mar 25 2026 11:59PM.

Sincerely,

Suzanne De Bruijn, PhD

Associate Editor

PLOS Medicine

sbruijn@plos.org

---

## [Editor Report · Decision Letter 4]

20 Mar 2026

Dear Dr Man,

On behalf of my colleagues and the Academic Editor, Seena Fazel, I am pleased to inform you that we have agreed to publish your manuscript "Association between gabapentinoid treatment, concurrent use with opioid or benzodiazepine and the risk of drug poisoning – a self-controlled case series study " (PMEDICINE-D-25-03057R4) in PLOS Medicine.

We have one last request: for the competing interest statement, we thank you for listing all COIs. However, can you add the following sentence at the end? "the other authors have declared that no competing interests exist".

Furthermore, before your manuscript can be formally accepted you will need to complete some formatting changes, which you will receive in a follow up email. Please be aware that it may take several days for you to receive this email; during this time no action is required by you. Once you have received these formatting requests, please note that your manuscript will not be scheduled for publication until you have made the required changes.

PRESS

Sincerely,

Suzanne De Bruijn, PhD

Associate Editor

PLOS Medicine